# Cumulative Ecological Risk and Academic Burnout in Chinese College Students: A Moderated Mediation Model

**DOI:** 10.3390/ijerph20031712

**Published:** 2023-01-17

**Authors:** Yan Liu, Yan Zhang, Cong Peng, Yaojin Li, Qianbao Tan

**Affiliations:** 1School of Education, Hunan University of Science and Technology, Xiangtan 411201, China; 2School of Education, Huazhong University of Science and Technology, Wuhan 430074, China

**Keywords:** cumulative ecological risk, neuroticism, academic burnout, college student

## Abstract

This study explored the relationship between cumulative ecological risk exposure and academic burnout among Chinese college students and the mediating and moderating effects of neuroticism and gender, respectively. A total of 580 college students were selected as participants. They completed a battery of questionnaires that measured cumulative ecological risk, neuroticism, and academic burnout. The results showed that: (1) cumulative ecological risk was positively related to neuroticism and academic burnout; (2) cumulative ecological risk positively predicted academic burnout; (3) neuroticism partly mediated the relationship between cumulative ecological risk and academic burnout; and (4) gender moderated the effect of cumulative ecological risk and academic burnout. A high level of cumulative ecological risk had a greater impact on neuroticism among women, compared to men. These findings advance our current knowledge of the specific effects of cumulative ecological risk on academic burnout and the underlying internal mechanisms of this relationship. Furthermore, this study provides a constructive perspective on preventing and reducing academic burnout among college students.

## 1. Introduction

Academic burnout refers to emotional exhaustion, academic alienation, and a low sense of achievement caused by students’ excessive learning needs [1,2]. Researchers have repeatedly found that academic burnout is a common problem among college students [3,4]. For example, a previous study found that 23.0% of Chinese college students experience academic burnout [5]. This reflects students’ negative psychological state and affects their mental health, academic performance, and interpersonal relationships [1]. Consequently, they exhaust their energy and lose enthusiasm for learning. For college students, academic burnout seriously impacts not only their academic achievement but also their mental health [6]. It can lead to several adverse outcomes, including school dropout, depression [7], and psychological and behavioral problems [8]. Given its growing prevalence and detrimental effects on mental health, academic burnout among Chinese college students demands persistent and concerted research efforts.

The cumulative risk model argues that individual developmental outcomes are better predicted by examining the accumulation of risk factors than by focusing on the adverse consequences of singular indicators [9,10]. Empirical studies have focused on the relationship between cumulative ecological risk and the academic competence and achievement of children and adolescents [11,12]. However, few studies have examined the relationship between cumulative ecological risk and academic burnout. Moreover, the bioecological model of human development [13,14] suggests the need to examine how cumulative ecological risk affects academic burnout among college students. Therefore, the primary purpose of this study is to investigate the mediating role of neuroticism between cumulative ecological risk and academic burnout among college students. In line with research calling for further investigation of moderators in the relationship between risk and individual outcomes [9], we also examined gender as a potential moderating factor. This study contributes significantly to our understanding of how to prevent and reduce academic burnout.

### 1.1. Cumulative Ecological Risk and Academic Burnout

Risk factors are individual or environmental factors associated with an increased likelihood of negative or undesirable outcomes [15]. Bronfenbrenner’s ecological theory [16] highlights the potential for risk in various social contexts (e.g., family, school, peers, and community). Several social contextual factors contribute to the risk of academic burnout. A large body of research has supported the cumulative risk model. Individuals who experience multiple risk factors early in life have more adverse developmental outcomes than those who experience singular risks [9,10,17]. Considering the co-occurrence of risk factors, researchers have applied a bioecological model of human development [13,14] to identify the consequences of multiple risk factors in college students’ environments. Most studies on exposure to multiple risk factors have relied on the cumulative risk model as a measure of multiple risks.

Academic burnout is also closely related to stress [18]. Burnout has most frequently been conceptualized as an individual’s affective response to the depletion of energy resources following exposure to chronic job stress [19]. Academic burnout is an important influencing factor for students’ academic progress and refers to a negative emotional, attitudinal, and behavioral response to stress arising from a failure to cope with academic pressure or solve learning problems [20]. If stressors are simultaneously present across several spheres, individuals may experience psychological discomfort, and their coping reserves may be taxed.

Empirical studies have shown that cumulative risk is negatively associated with academic competence and achievement among children and adolescents [11,12]. Previous research has shown that cumulative school and neighborhood risk factors, as well as their interactions, are significantly associated with school-wide achievement [21]. In particular, it was found that, as the number of risk factors increased, adolescents’ academic achievement decreased. In addition, adolescents with a lower level of academic achievement have been shown to experience higher levels of burnout than their peers who perform better in school [22]. From this perspective, high-risk college students may have poorer academic performance than low-risk ones. Individuals who grow up in environments with multiple risk factors are often unable to receive emotional support from their families, schools, or peers. Faced with learning pressure, they lack resources that can help them cope, resulting in lower academic achievement. Over time, college students can exhaust their resources and lose enthusiasm for learning. Thus, college students exposed to multiple risk factors are more likely to experience academic burnout. Therefore, based on the above, we hypothesized the following:

**Hypothesis** **1.**
*Cumulative ecological risk positively predicts academic burnout among college students.*


### 1.2. Mediating Role of Neuroticism

Neuroticism is a broad domain of negative affect that includes predispositions toward experiencing distressing emotions such as anxiety, anger, depression, and shame [23]. Accumulating evidence indicates that long-standing environmental conditions may be linked to the longitudinal trajectories of neuroticism [24]. Studies have found that adverse or stressful life events can affect neuroticism [24,25,26]. For example, in previous research, participants who reported experiencing an extremely stressful event or stressful life events showed higher neuroticism than participants who did not experience these types of events [24,25,26]. Notably, individuals with multiple risk factors experience more adverse or stressful life events, thereby experiencing increased neuroticism.

Neuroticism is a powerful predictor of academic burnout. As a disposition toward interpreting events negatively, neuroticism may adversely influence burnout levels over time [27]. Based on conservation of resources (COR) theory, neuroticism consumes resources and is likely to lead to resource burnout because the higher the level of neuroticism, the more pronounced the tendency to view the world pessimistically and interpret many stimuli as threatening [28]. Previous research has shown that neuroticism is positively correlated with, and can positively predict, academic burnout [29].

This study extends previous research by examining the mediating role of neuroticism in the relationship between cumulative ecological risk and academic burnout. Individuals with multiple risk factors may have a high level of neuroticism. They may experience adverse or stressful life events and, consequently, increased neuroticism. Moreover, individuals with high neuroticism tended to focus more on the negative features of stressful events. This negative disposition consumes resources, which can lead to academic burnout. Based on the above, we hypothesize the following:

**Hypothesis** **2.**
*Neuroticism mediates the relationship between cumulative ecological risk and academic burnout.*


### 1.3. Moderating Role of Gender

Evans et al. [9] highlighted the need to explore potential moderators of cumulative ecological risk. In addition to the mediation model, we examined the moderating role of gender in the relationship between cumulative ecological risk and neuroticism. First, according to the diathesis–stress model [30], different types of stressors or risk factors may be perceived differently by men and women. Given the gender socialization processes that promote the centrality of social relationships for girls and the tendency among girls to construct their self-image using cues from others [31], they might be more acutely aware of their social location. Meanwhile, a large body of literature has highlighted girls’ sensitivity to interpersonal stressors in the development of internalization [32]. For example, Sullivan et al. [33] found that girls were at higher risk of problems in the context of high cumulative family socioeconomic status (SES) risk. Studies have also found that familial risk factors, such as parental depression, may affect boys and girls differentially [34]. Second, previous studies examining the impact of childhood risk on educational attainment or maladjustment have often identified significant gender differences [35,36]. For example, Horan and Widom [36] found that gender interacts with cumulative risk to influence educational attainment. Third, gender differences in neuroticism have been consistently reported, with women tending to score higher than men on measures of neuroticism [37,38]. Based on these findings, we hypothesized the following:

**Hypothesis** **3.**
*Gender moderates the relationship between cumulative ecological risk and neuroticism.*


In summary, we tested the following hypotheses: (a) cumulative ecological risk is positively correlated with academic burnout; (b) neuroticism mediates the relationship between cumulative ecological risk and academic burnout; and (c) gender moderates the relationship between cumulative ecological risk and neuroticism. Figure 1 illustrates the proposed model.

## 2. Methods

### 2.1. Participants and Procedure

A total of 580 college students (*M_age_* = 18.30, *SD* = 1.33) from four universities in Hunan Province, China, participated in this survey. Using random cluster sampling, we recruited participants by grade and class. Six hundred paper questionnaires were distributed, and 580 valid questionnaires were collected, with an effective rate of 96.67%. The student sample included 327 men (56.4%) and 253 women (43.6%). Of these, 343 (59.1%) and 237 (40.9%) were freshmen and sophomores, respectively. The questionnaires were administered to the students in classroom settings by their teachers. Informed consent was obtained from all participants. Participants completed the questionnaires anonymously. We highlighted student confidentiality by assuring the students that no teachers or schools could access their personal data. The study procedures were approved by our university’s Ethics Committee of the Institution of Social Science Surveys.

### 2.2. Cumulative Ecological Risk Analysis and Measures

According to Bronfenbrenner’s (1989) [16] ecological theory, risk factors were chosen from four social domains in this study: family, peers, school, and neighborhood. The selection of risk factors was guided by two sources: research documenting the major correlates of academic burnout and variables commonly employed in studies of multiple risk exposures. Thus, we selected 10 risk factors to examine the relationship between cumulative ecological risk and academic burnout among Chinese college students.

Within the familial sphere, low SES is an important individual risk factor for academic achievement and burnout [39,40,41]. According to the family stress model, low SES results in practical difficulties for families, which creates an environment that impairs the mental health of individuals in the family [39,42] and can negatively affect parental rearing behavior [43], such as by leading to harsh parenting [44]. Harsh parenting can have negative effects on individual academic development [45] and achievement [46]. Researchers have also found that alienation from one’s parents can predict academic burnout [40]. Therefore, we selected low family SES, harsh parenting, alienation from one’s mother, and alienation from one’s father as familial risk factors.

Previous studies have found that low peer support and alienation are risk factors for academic burnout [21,40]. A meta-analysis showed that peer support can impact academic burnout [47]. Thus, we selected alienation from peers and low peer support as risk factors for academic burnout.

Further, past research has identified school and neighborhood or community as risk factors related to achievement [21,48]. Specifically, low school connectedness [49] and community safety [48,50] were risk factors selected for this study.

Additionally, we selected left-behind experiences and foster placements as risk factors. Left-behind experiences during childhood have been recognized as a public health problem in China [51] and can have sustained effects on university students in early adulthood [52]. Regarding foster placement, Rutter (1983) found that it is significantly correlated with individual development [53].

In conclusion, based on the findings of previous empirical studies conducted on college students, this study created cumulative risk indices. Ten variables were selected: low family SES, harsh parenting, alienation from one’s mother, alienation from one’s father, low school connectedness, alienation from peers, low peer support, low community safety, left-behind experience, and foster placement.

Rutter [53] proposed a measurement model for cumulative risk. The cumulative risk model is parsimonious, statistically sensitive, and fits well with underlying theoretical models. This model was constructed by dichotomizing each risk factor exposure (0 = no risk, 1 = risk), and then summing the dichotomous scores.

#### 2.2.1. Family Socioeconomic Status

Consensus indicates that income, education, and occupation together represent SES better than any of these factors alone [39,54]. We measured family SES using three indicators assessing parents’ educational attainment, current occupation, and monthly family income [40,55]. Parental education was divided into three levels, scored from 1 to 3; occupations were divided into eight levels, scored from 1 to 8; and monthly family income was divided into five levels, scored from 1 to 5. Higher scores indicated higher family SES.

#### 2.2.2. Harsh Parenting

Harsh parenting was assessed using the Chinese version of the Harsh Parenting Inventory [46]. The inventory consists of four items rated on a five-point scale (1 = never, 5 = always). Higher scores indicated higher levels of harsh parenting. In this study, the Cronbach’s α was 0.79.

#### 2.2.3. Alienation from One’s Mother, Father, and Peers

We used the subscales of the Chinese version of the Revised Inventory of Parent and Peer Attachment (IPPA-R) [56,57] to assess college students’ alienation from their parents and peers. The subscale of alienation from one’s mother has six items, and Cronbach’s α was 0.70 in this study. The subscale of alienation from one’s father has six items, and the Cronbach’s α was 0.71. The subscale of alienation from peers has seven items, and the Cronbach’s α was 0.72. The items are rated on a five-point Likert scale (1 = strongly disagree, 5 = strongly agree). Higher scores indicated greater alienation from parents and peers.

#### 2.2.4. School Connectedness

The School Connectedness Scale was used to measure college students’ school connectedness [58,59]. This scale contains six items, including teacher support and school belonging. The items are rated on a five-point Likert scale (1 = strongly disagree, 5 = strongly agree). Higher scores indicated higher school connectedness. In this study, the Cronbach’s α was 0.80.

#### 2.2.5. Peer Support

Peer support was measured using a subscale of the Chinese version of the Healthy Kids Resilience Assessment (HKRA) [48,60]. This subscale has three items that reflect peer support. Participants were required to indicate the degree to which the items described them on a four-point scale (1 = strongly disagree, 4 = strongly agree). An example item is “When I’m in trouble, my friends can help me”. Higher scores indicated higher levels of peer support. In this study, the Cronbach’s α was 0.79.

#### 2.2.6. Community Safety

Community safety was assessed using one item [35] that asked college students to rate the safety of their community on a four-point scale (1 = very unsafe, 4 = very safe).

#### 2.2.7. Left-Behind Experience

Left-behind experience was defined as parent–child separation for at least six months. It was measured using one item [61] that asked college students whether they had more than six months of left-behind experience. If so, left-behind experience was scored as 1; if not, the item was scored as 1.

#### 2.2.8. Foster Placement

Foster placement was assessed using a single item asking participants whether they had foster care experience. A “no” was counted as 0, and “yes” as 1.

### 2.3. Neuroticism

Neuroticism was measured using a subscale of the Chinese version of the Revised NEO-Personality Inventory (NEO-PI-R) [21,62]. This subscale comprises 48 items rated on a five-point scale (1 = strongly disagree, 5 = strongly agree). Higher scores indicate higher neuroticism. In this study, the Cronbach’s α was 0.76.

### 2.4. Academic Burnout

Academic burnout was measured using the Academic Burnout Scale [1], which includes 20 items rated on a five-point scale (1 = strongly disagree, 5 = strongly agree). Higher scores indicate higher levels of academic burnout. In this study, the Cronbach’s α was 0.85.

### 2.5. Data Analysis

We used SPSS version 23.0 for descriptive statistics with an independent sample *t*-test and correlation analysis. We computed the mean, standard deviation, and correlation coefficient for each variable to describe the data. Then, we ran the PROCESS version 3.3 macro program [63] to test mediating and moderating effects. First, we used Model 4 (a simple mediating model) in PROCESS for SPSS to test the mediating effect of neuroticism on cumulative ecological risk and academic burnout. Second, we used Model 7 (assuming the first half path of the mediating model was moderated, consistent with the proposed model of this study) in PROCESS to examine the moderated mediating effect of cumulative ecological risk on academic burnout.

## 3. Results

### 3.1. Description of Risk Indicators

Risk was calculated as a simple summation of singular risk factors. Each risk factor was coded dichotomously to indicate its absence (0) or presence (1). Table 1 provides the descriptive information for each risk variable. Subsequently, the number of risk factors was tallied across the participants, yielding a cumulative risk score. The scores ranged from 0 to 10, with higher scores indicating greater risk.

### 3.2. Preliminary Analysis

Independent sample *t*-tests were conducted using gender (1 = male; 0 = female) and grade (1 = freshman; 2 = sophomore) as independent variables and academic burnout as the dependent variable. The results showed no significant differences in academic burnout among college students based on gender (*t* = 1.49, *p* > 0.05) or grade (*t* = 1.01, *p* > 0.05). However, significant gender differences were observed for neuroticism (*t* = 3.79, *p* < 0.001; *M_male_* = 140.50, *SD* = 12.10; *M_female_* = 144.74, *SD* = 14.85).

Descriptive statistics (mean and standard deviation) and Pearson’s correlation coefficients for the study variables are presented in Table 2. Correlation analysis showed that cumulative ecological risk was positively associated with neuroticism (*r* = 0.42, *p* < 0.001) and academic burnout (*r* = 0.30, *p* < 0.001). Neuroticism was positively associated with academic burnout (*r* = 0.40, *p* < 0.001).

### 3.3. Mediation Analysis

The Hayes [63] PROCESS macro was used to examine the mediating role of neuroticism in the relationship between cumulative ecological risk and academic burnout. After controlling for grade, the regression equation was significant (*R* = 0.30, *R*^2^ = 0.09, *F*_(2,577)_ = 28.62, *p* < 0.001). Cumulative ecological risk significantly predicted academic burnout (*β* = 0.30, *t* = 7.49, *p* < 0.001, 95% CI = [0.22, 0.38]), thus supporting Hypothesis 1.

Using neuroticism as a mediating variable, the regression equation was significant (*R* = 0.43, *R*^2^ = 0.19, *F*_(3,576)_ = 44.26, *p* < 0.001). Cumulative ecological risk (*β* = 0.15, *t* = 3.73, *p* < 0.001, 95% CI = [0.07, 0.24]) and neuroticism (*β* = 0.36, *t* = 8.29, *p* < 0.001, 95% CI = [0.27, 0.44]) also significantly predicted academic burnout. The bias-corrected bootstrap method showed that the indirect effect of neuroticism on academic burnout was significant, with the index of the mediation effect being 0.14 (95% CI = [0.10, 0.19]). Mediation accounted for 48.27% of the total effect of academic burnout. The direct effect (0.15) of cumulative ecological risk on academic burnout was also significant (95% CI = [0.07, 0.24], *t* = 3.73, *p* < 0.001). Therefore, neuroticism partially mediated the relationship between cumulative ecological risk and academic burnout, thus supporting Hypothesis 2.

### 3.4. Moderated Mediation Analysis

Table 3 presents the main results of the moderated mediation analysis using Hayes (2013) [63] PROCESS macro (Model 7). Cumulative ecological risk had a significant effect on neuroticism (*β* = 0.39, *t* = 10.94, *p* < 0.001, 95% CI = [0.32, 0.47]), and this effect was moderated by gender (*β* = −0.16, *t* = −2.29, *p* < 0.05, 95% CI = [−0.30, −0.02]), thus supporting Hypothesis 3. The results were displayed in Figure 2.

To further reveal the nature of this moderation, we performed a simple slope analysis. As shown in Figure 3, high CR was represented by 1 SD above the mean, and low CR was represented by 1 SD below the mean. Simple slope tests indicated that high CR has a greater impact on neuroticism in women (*simple slope* = 0.49, *t* = 9.53, *p* < 0.001, 95% CI = [0.39, 0.59]) than in men (*simple slope* = 0.32, *t* = 6.43, *p* < 0.001, 95% CI = [0.22, 0.42]).

## 4. Discussion

This study aimed to improve the current understanding of the relationship between exposure to an accumulation of risk factors and academic burnout among Chinese college students. The results showed that college students who are exposed to multiple risk factors are much more likely to develop academic burnout than their peers without such risk exposure. Thus, our findings extend the existing research by providing a clearer description of academic burnout among Chinese college students. Moreover, these findings provide an accurate overview of the mediating role of neuroticism and the moderating role of gender in the relationship between cumulative ecological risk and academic burnout.

### 4.1. Cumulative Ecological Risk and Academic Burnout

The present research shows that cumulative ecological risk is significantly and positively correlated with academic burnout, with findings indicating that an accumulation of environmental risk factors is associated with a linear increase in academic burnout. Our findings are consistent with the bioecological model of human development [13,14], which posits that the social contexts in which college students interact are crucial determinants of academic burnout. This is consistent with previous research, which clarified that cumulative ecological risk is negatively correlated with school-wide achievement [43] and academic burnout [34]. Chronic or repeated exposure to risks can result in increased “wear and tear” of the stress system, and may increase the risk of academic burnout [64]. In this study, a context characterized by low family SES, harsh parenting, alienation from parents and peers, low school connectedness, peer support, low community safety, left-behind experiences, and foster placement might create an unpredictable and stressful environment for college students that facilitates the development of academic burnout. One possible explanation is that cumulative risk is negatively associated with academic competence and achievement [11,12,21]. Moreover, college students with lower academic achievement experience higher levels of burnout than those who perform better in school [37].

### 4.2. Mediating Role of Neuroticism

The findings from the hypothesized pathway support the hypothesis that neuroticism serves as a mediator linking cumulative ecological risk with academic burnout. This finding is consistent with previous theoretical assumptions [9,65]. We adopted the COR theory to explain burnout. Specifically, cumulative ecological risk can predict academic burnout. According to COR theory, stress results either from the threatened or actual loss of resources or from not receiving enough gain in resources from an investment in other resources [65]. College students with a high cumulative ecological risk experience adverse or stressful life events and resource loss. Neuroticism acts as a negative filter in everyday life and magnifies the impact of adverse events [66], which aggravates the loss of resources. College students in stressful environments may not have sufficient social support from their parents or peers to obtain a high level of academic achievement and could therefore easily experience burnout.

This study’s findings are consistent with those of previous research on the relationships between cumulative ecological risk, neuroticism, and academic burnout [27,28,29]. Several potential explanations exist for neuroticism’s role as an underlying mechanism in this context. First, cumulative ecological risk can increase the risk of neuroticism. College students who report multiple risk factors in their environment encounter more adverse or stressful life events and experience increased neuroticism. Cumulative ecological risk factors reflect a lack of supportive resources. If they have several disadvantages in the family, peer, school, and community domains, college students may have difficulty handling stressful events and experience setbacks. According to previous studies, long-term environmental conditions, with adverse or stressful life events, have been linked to neuroticism [24,25,26]. Second, cumulative ecological risk can increase the risk of academic burnout in individuals with high levels of neuroticism. Neuroticism is a well-known stress vulnerability factor strongly associated with burnout [27,67]. Currently, higher education students are experiencing greater stress levels than ever before in this demanding knowledge-based society [68]. Research indicates that individuals with high neuroticism levels may react more negatively to stressful events than those with lower levels [69]. Under stress, they tend to focus on the negative elements of the situation and retreat from challenges [70]. Higher education systems worldwide have origins in different cultural backgrounds. In China, the learning environment is highly competitive and emphasizes the importance of education [71]. Researchers have found that academic burnout is a common problem among college students in China [3,4,5]. College students with high neuroticism experience frequent negative emotions and burnout, especially in the context of strong education competition.

### 4.3. Moderating Role of Gender

This study shows that gender plays a moderating role in the relationship between cumulative ecological risk and neuroticism, supporting Hypothesis 3. Specifically, a high level of cumulative ecological risk has a greater impact on neuroticism in women than in men. This finding aligns with those of previous studies [32,33]. There are two potential explanations for this observation.

First, the present study used both familial (low family SES, harsh parenting, and alienation from father and mother) and interpersonal (alienation from peers and peer support) risk factors, which women might perceive sensitively. According to the diathesis-stress model [30], women possess vulnerabilities (i.e., diatheses) to maladaptive outcomes that assert themselves in the context of certain environments. Studies have suggested that women are more vulnerable to environmental and familial stressors [34]. Women who experience internal stressors [72] are more vulnerable to subsequent internalizing problems, especially when they are from families with multiple SES-related stressors [32,33]. Thus, a low SES environment could potentially exacerbate women’s tendencies toward developing internalizing problems. In this study, high-risk environments such as SES disadvantages and alienation from parents and peers might have been deeply internalized by female participants, leading to increased neuroticism.

Second, compared with men, women in high-risk environments are more sensitive to emotional stimuli. Neurophysiological studies have shown significant differences in adrenal function, pituitary hormone levels, and hypothalamic function between men and women under stress [73]. The activity of the dopamine system in the prefrontal cortex of women is higher than that of men, resulting in abnormal sensitivity of the socioemotional system in the brain to various emotional stimuli [73]. Therefore, the cumulative ecological risk of neuroticism is more pronounced in women than in men.

### 4.4. Limitations and Implications

The results of this study should be interpreted within the context of several limitations. First, we used a cross-sectional research design; thus, causal effects of the model cannot be inferred. Second, our participants were sampled from a Chinese cultural context. Higher education systems differ based on cultural background, and our findings may not be generalizable to other regions. Third, this study adopted a self-report method for data collection, and participants’ responses may reflect some bias [74], which can lead to common method bias, create shared variance, and involve social desirability. Fourth, some limitations are commonly found in cumulative ecological risk studies [9], including limitations in the selection of risk variables and an inability to examine dose–response relationships given the reliance on dichotomous risk factors. Additionally, the sample size was insufficient for this study. Future research should increase the sample size to verify the relationship between cumulative ecological risk and academic burnout.

Future studies could use longitudinal and experimental methods to further examine the mediating role of neuroticism in the relationship between cumulative ecological risk and academic burnout. Cross-cultural studies should be conducted in the future to compare the effects of different cultures. Future researchers should include objective measures or multiple reports to improve the accuracy of the variables.

Despite these limitations, this study makes an important contribution to the literature on cumulative ecological risk and academic burnout. First, the results revealed a significantly negative association between cumulative ecological risk and academic burnout among Chinese college students, indicating that those with a high level of cumulative ecological risk are more likely to experience academic burnout. Second, we found that neuroticism mediates the association between cumulative ecological risk and academic burnout, indicating the importance of neuroticism in decreasing academic burnout among college students. These findings enrich the current understanding of the potential mechanism of the impact of cumulative ecological risk on academic burnout among college students. Future research can extend this line of investigation by uncovering the psychosocial processes underlying the cumulative effects. Third, we constructed a moderated mediation model and found that gender moderated the relationship between cumulative ecological risk and neuroticism. Interventions targeted to help Chinese college students minimize academic burnout must reduce the impact of multiple risk variables and neuroticism. Furthermore, it would be advantageous to tailor interventions to students experiencing many risk factors and students with neuroticism. Girls are particularly susceptible to neuroticism when risk levels are high, highlighting the importance of paying special attention to reducing their cumulative ecological risk.

## 5. Conclusions

The current study investigated the mediating role of neuroticism between cumulative ecological risk and academic burnout and examined gender as a moderating factor among Chinese college students. These findings confirm the cumulative effects of risk factors on academic burnout among college students. Although exposure to multiple risks can be challenging for college students facing academic burnout, our results provide a potential way to understand its effect paths and group differences, thus expanding and deepening research on the mechanisms underlying college students’ academic burnout. Academic burnout is a common problem among college students in China. These promising results indicate that the application of the bioecological model of human development and the cumulative ecological risk model can provide a useful guide for future research on Chinese college students’ academic burnout.

## Figures and Tables

**Figure 1 ijerph-20-01712-f001:**
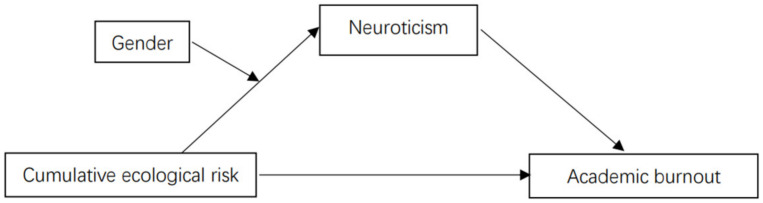
The proposed moderated mediation model.

**Figure 2 ijerph-20-01712-f002:**
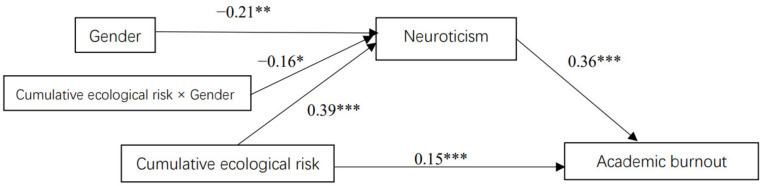
The moderated mediation model (male = 1, female = 0). * *p* < 0.05, ** *p* < 0.01, *** *p* < 0.001.

**Figure 3 ijerph-20-01712-f003:**
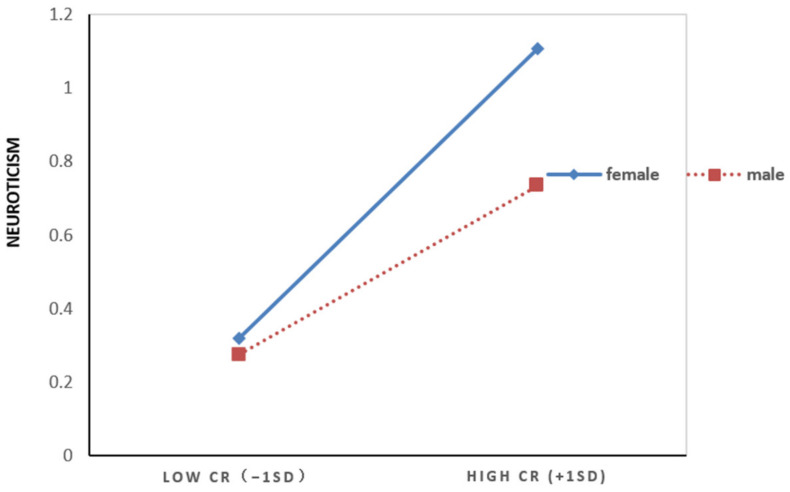
Moderating effect of gender in the relationship between cumulative ecological risk and neuroticism. Note. CR: cumulative ecological risk.

**Table 1 ijerph-20-01712-t001:** Description of risk indicators and the cumulative risk index.

Risk Variable	Items	Risk Status Criterion	At-Risk (%)
Family socioeconomic status	5	≤25th percentile	28.8
Harsh parenting	4	≥75th percentile	31.2
Alienation from mother	6	≥75th percentile	27.8
Alienation from father	6	≥75th percentile	29.3
School connectedness	6	≤25th percentile	25.0
Alienation from peers	7	≥75th percentile	28.3
Peer support	3	≤25th percentile	30.7
Community safety	1	Students who responded “very unsafe” or “unsafe” to the item asking if they feel safe in their community.	42.1
Left-behind experience	1	Students who responded yes to the item asking if they have > 6 months left-behind experience.	18.8
Foster experience	1	Students who responded yes to the item asking if they experienced foster placement.	3.4

**Table 2 ijerph-20-01712-t002:** Correlations of the main study variables (*N* = 580).

Variables	*M*	*SD*	1	2	3
1. Cumulative ecological risk	2.65	1.92	1		
2. Neuroticism	142.35	13.52	0.42 ***	1	
3. Academic burnout	55.94	10.63	0.30 ***	0.40 ***	1

Note: *** *p* < 0.001.

**Table 3 ijerph-20-01712-t003:** Testing the moderated mediation effect of cumulative ecological risk on academic burnout.

Regression Equation (*N* = 580)		Fit Indicators			Coefficient Significance	
Outcomes	Predictors	*R*	*R* ^2^	*F*	*β*	*t*
Neuroticism		0.52	0.27	52.22 ***		
	Grade				−0.50	−6.69 ***
	Cumulative ecological risk				0.39	10.94 ***
	Gender				−0.21	−2.79 **
	Cumulative ecological risk × Gender				−0.16	−2.29 *
Academic burnout		0.43	0.19	44.26 ***		
	Grade				0.16	1.96
	Cumulative ecological risk				0.15	3.73 ***
	Neuroticism				0.36	8.29 ***

Note: * *p* < 0.05, ** *p* < 0.01, *** *p* < 0.001; gender is a binary variable (1 = male, 0 = female).

## Data Availability

Data available on request from the authors.

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
