# Peer review of "Cumulative Ecological Risk and Academic Burnout in Chinese College Students: A Moderated Mediation Model"

_ijerph, 2023, doi:10.3390/ijerph20031712_

Round 1
Reviewer 1 Report
Dear editor,
Thanks for giving me the chance to review this article. My comments are as follows.
1. “Cumulative risk” seems not to be a suitable description for cross-sectional design. What kinds of risk you mean? It is unclear.
2. The part of 1.1 is very confusing. If you would like to state how you chose the covariates, you should put this part in the methods. If you wanted to clarify some theories about you model, you should give more clear information.
3. In the epidemiological study, cumulative risk is standard rate for survival analysis. Although cumulative risk might belong to a model, you should choose another words to describe it and give a more detailed information for the “risk” (What kinds of risk?).
4. What do you use for sampling? It is very confusing if you only stated 580 college students included in this study.
5. The part of data analysis is too simple.
6. If you chose to use percentile to describe your included variables, mean and sd are not suitable for statistical description. Do all variables conform to the normal distribution?
7. Basic estimates from regression models are needed.
8. A figure of paths along with parameters is more useful when performing your models.
9. The sample size is not enough for your analysis, which could be put in the part of limitations.
10. The conclusions are with repetition of the results. Please rewrite it.
Author Response
Dear Editors and reviewers:
Thank you for your precious comments and advice. Those comments are all valuable and very helpful for revising and improving our paper, as well as the important guiding significance to our researches. We have studied comments carefully and have made correction which we hope meet with approval. Revised portion are marked in the paper. The main corrections in the paper and the responds to the reviewers’ comments are in the attachment. Please download the attachment. Thank you very much.

Reviewer 2 Report
Relevant variables of focus to students, particularly with burnout and anxiety being heightened by the COVID lockdown and return to in-person instruction. The application of the cumulative risk model was expertly applied. Specific computed statistics were appropriately analyzed and reported and interpreted in table form. Implications were derived and solidly grounded in the findings. I would like to see some recommendations for practical application and future related research to build upon these findings. An outstanding research study that fully merits publication!
Author Response
Dear Editors and reviewers:
Thank you for your precious comments and advice. Those comments are all valuable and very helpful for revising and improving our paper, as well as the important guiding significance to our researches. We have studied comments carefully and have made correction which we hope meet with approval. Revised portion are marked in the paper. The main corrections in the paper and the responds to the reviewers’ comments are as flowing and in the attachment.
Q1:Relevant variables of focus to students, particularly with burnout and anxiety being heightened by the COVID lockdown and return to in-person instruction. The application of the cumulative risk model was expertly applied. Specific computed statistics were appropriately analyzed and reported and interpreted in table form. Implications were derived and solidly grounded in the findings. I would like to see some recommendations for practical application and future related research to build upon these findings. An outstanding research study that fully merits publication!
Response:We appreciate your positive evaluation of our work. Those comments are valuable and helpful for revising and improving our paper. We have revised the manuscript accordingly. Our point-by-point responses are detailed below.
First, we add some recommendations for practical application. Please see as follows or page 12 in the manuscript.
“Interventions targeted to help Chinese college students minimize academic burnout must reduce the impact of multiple risk variables and neuroticism. Furthermore, it would be advantageous to tailor interventions to students experiencing many risk factors and students with neuroticism. Girls are particularly susceptible to neuroticism when risk levels are high, highlighting the importance of paying special attention to reducing their cumulative ecological risk.”
Second, we added future related research building upon our findings. Please see as follows or page 11-12.
“Future studies could use longitudinal and experimental methods to further examine the mediating role of neuroticism in the relationship between cumulative ecological risk and academic burnout. Cross-cultural studies should be conducted in the future to compare the effects of different cultures. Future researchers should include objective measures or multiple reports to improve the accuracy of the variables.”

Reviewer 3 Report
Originality: excellent, contribution to the field: good, clarity of presentation: good, depth of research: good
The results presented by the Authors are interesting and important. Author(s) provides original results of their investigations and examination of material from their own collections. The applied methods and the interpretation and presentation of results correspond to international standards. Methodology of the research is clear. The aims, background of the research problem, hypothesis and research methodology are clearly described. Findings are useful for educational, pedagogical, social environments. Author(s) have studied and used an appropriate number of bibliography sources. The language is not always perfect, the syntax is in some parts a bit convoluted.
Author Response
Dear Editors and reviewers:
Thank you for your precious comments and advice. Those comments are all valuable and very helpful for revising and improving our paper, as well as the important guiding significance to our researches. We have studied comments carefully and have made correction which we hope meet with approval. Revised portion are marked in the paper. The main corrections in the paper and the responds to the reviewers’ comments are as flowing and in the attachment.
Reviewer3
Thank you for your suggestion. We really appreciate your efforts in reviewing our manuscript. We have revised the manuscript accordingly.
The results presented by the Authors are interesting and important. Author(s) provides original results of their investigations and examination of material from their own collections. The applied methods and the interpretation and presentation of results correspond to international standards. Methodology of the research is clear. The aims, background of the research problem, hypothesis and research methodology are clearly described. Findings are useful for educational, pedagogical, social environments. Author(s) have studied and used an appropriate number of bibliography sources. The language is not always perfect, the syntax is in some parts a bit convoluted.
Response:We appreciate your positive evaluation of our work. We apologize for the language problems in the original manuscript. We carefully checked the syntax and the language presentation was improved with assistance from a native English speaker with appropriate research background. We hope it can meet the journal’s standard. Thank you so much for your useful comments.

Round 2
Reviewer 1 Report
1. In the revision, the authors stated use " random cluster sampling". What is the cluster in this study?
2. Q8 in previous revision has some additional questions. The authors could not use "gender", "cumulative ecological risk" et al in the study. For example, the study should point out what gender, higher risk or lower risk et al in the figures and other places of the manuscript.
Author Response
Dear Editors and reviewers:
Thank you for your precious comments and advice. Those comments are all valuable and very helpful for revising and improving our paper, as well as the important guiding significance to our researches. We have studied comments carefully and have made correction which we hope meet with approval. Revised portion are marked in the paper. The main corrections in the paper and the responds to the reviewers’ comments are as flowing:
Response to the reviewer's comments:
Reviewer 1 (round 2)
Thank you for your suggestion. We really appreciate your efforts in reviewing our manuscript. We have revised the manuscript accordingly. Our point-by-point responses are detailed below and in the attachment. Please download the attachment. Thank you.
Q1. In the revision, the authors stated use " random cluster sampling". What is the cluster in this study?
Response:Special thanks for your comments and suggestions. Using random cluster sampling, we recruited participants by grade and class. Please see as follows or Line 146-147 in red font, underline.
“A total of 580 college students (Mage = 18.30, SD = 1.33) from four universities in Hunan Province, China participated in this survey. Using random cluster sampling, we recruited participants by grade and class.”
Q2. Q8 in previous revision has some additional questions. The authors could not use "gender", "cumulative ecological risk" et al in the study. For example, the study should point out what gender, higher risk or lower risk et al in the figures and other places of the manuscript.
Response:Thank you for your reminder. We revised the manuscript to point out what gender, higher risk or lower risk et al in the figures and other places. First, we added “(male = 1, female = 0)” to make it clear. Please see Line 306-307. Second, we revised the statement of simple effect. Please see as follows or Line 16, 309-318 and 387-388 in red font, underline.
“To further reveal the nature of this moderation, we performed a simple slope analysis. As shown in Figure. 3, High CR was represented by 1 SD above the mean and low CR was represented by 1 SD below the mean. Simple slope tests indicated that high CR has a greater impact on neuroticism in women (simple slope = 0.49, t = 9.53, p < 0.001, 95% CI = [0.39, 0.59]) than in men (simple slope = 0.32, t = 6.43, p < 0.001, 95% CI = [0.22, 0.42]).”
